# The Association of Dietary Magnesium and Prediabetes in Childbearing Chinese Women: Results from China Nutrition and Health Surveillance (2015–2017)

**DOI:** 10.3390/nu14214580

**Published:** 2022-11-01

**Authors:** Jingxin Yang, Huidi Zhang, Yang Cao, Xiaoyun Shan, Jie Feng, Jiaxi Lu, Shuya Cai, Liyun Zhao, Lichen Yang

**Affiliations:** National Institute for Nutrition and Health, Chinese Center for Disease Control and Prevention, Key Laboratory of Trace Element Nutrition, National Health Commission of the People’s Republic of China, Beijing 100050, China

**Keywords:** prediabetes, magnesium, dose-response effect, threshold

## Abstract

Although several studies have shown the relationship between magnesium and diabetes, there are few studies regarding the association between magnesium status and prediabetes. This study aimed to examine the association between dietary magnesium intake and the risk of prediabetes in childbearing Chinese women (18–44 years). A total of 1981 participants without diabetes were randomly selected from China Nutrition and Health Surveillance (CNHS) in 2015 by considering the regional types and monitoring points, including 1420 normal fasting glucose (NFG) people and 561 prediabetes cases. The Food Frequency Questionnaire (FFQ) and inductively coupled plasma mass spectrometry (ICP-MS) measured dietary and plasma magnesium. The results of this study showed that dietary magnesium intake was inversely associated with fasting plasma glucose. For every 100 mg of magnesium intake, there was a 0.052 mmol/L (95% CI 0.019, 0.085) reduction in fasting plasma glucose (*p* = 0.002). Compared with the lowest intake quartile (<350.10 mg/day), the third and fourth quartiles (≥405.41 mg/day) reduced the odds risk of prediabetes (OR = 0.63, 95% CI 0.46, 0.88, *p* = 0.020) by approximately 37%. The dose-response curves for the association between magnesium intake and prediabetes followed an L shape. The odds ratio of prediabetes decreased significantly with increasing dietary Mg intake at less than 410 mg/day, and then the curve leveled off or slightly increased. This study found a negative association between dietary magnesium intake and prediabetes in childbearing Chinese women. The risk of prediabetes was reduced with increasing dietary magnesium intake, and the threshold value was 410 mg/day. This suggests that childbearing women also need to pay attention to their magnesium status.

## 1. Introduction

Globally, the prevalence of type 2 diabetes (T2DM) is high and rising across all regions. The prevention and control of T2DM remain a public priority [1]. The prevalence of prediabetes is also increasing worldwide and it is projected that >470 million people will have prediabetes in 2030 [2]. Prediabetes represents a massive reservoir of patients at risk of diabetes. Their detection opens the door to interventions that can lead to the prevention of T2DM [3]. If the intervention is appropriate, 5–10% of people can go back to normoglycaemia. Magnesium (Mg) is an essential mineral for the human body, which plays a significant role in glucose and insulin metabolism [4]. A recent meta-analysis [5] showed that circulating Mg levels in prediabetes were significantly lower than in healthy controls, confirming that Mg deficiency may play a role in the development and progression of prediabetes [6,7,8]. Dietary Mg is the main source of human Mg, which directly affects the Mg nutritional status of the body [9]. GUERRERO-ROMERO’s RCT study [10] included 116 men and non-pregnant women who were newly diagnosed with prediabetes. The supplement group received 30 mL of MgCl2 5% solution once daily for four months. The results demonstrated that plasma glucose levels and glycemic status in prediabetes were improved by oral Mg supplementation.

Studies on prediabetes and dietary Mg intake are scarce, especially in childbearing women, and the vast majority of research was about the relationship between dietary Mg intake, metabolic syndrome (Mets) [11], coronary heart disease (CHD) [12], and type 2 diabetes mellitus (T2DM) [13,14]. ADELA-HRUBY’s cohort research [8] also certified that women were at higher risk for low dietary magnesium intake than men.

Therefore, we aimed to explore the association between dietary Mg intake and prediabetes in childbearing Chinese women (18–44 years).

## 2. Materials and Methods

### 2.1. Subjects

The subjects were from China Nutrition and Health Surveillance (2015) (CNHS 2015). Sampling design information is stated elsewhere [15]. Stratified random sampling was used to select the required samples from 18–44-year-old women of childbearing age (excluding pregnant women) for analysis. The sample size calculating formula of the cross-sectional study was N = $deff\frac{u^{2}p(1-p)}{d^{2}}$. According to LIMIN-WANG’s research, the prevalence of prediabetes in Chinese women in 2013 was 35.0% [1]. The values of the parameters were *u* of 1.96, *p* of 0.35, *deff* of 1.5, and *d* of 4%. Considering the sample stratification (city/urban), the minimum sample size was 1640 people. To meet both the minimum sample size and be nationally representative, eight people were taken from each monitoring site, considering two factors: age and urban/rural residence. In short, 2312 people were selected from 289 monitoring sites. Then, 59 individuals were excluded due to missing biochemical and basal measurements. To ensure the accuracy of the dietary survey, we deleted 272 individuals whose energy intake per day was out of the 800–4000 Kcal range after being calculated by the Food Frequency Questionnaire (FFQ). Finally, a total of 1981 women of childbearing age were included (Figure 1). The protocol has been approved by the Ethics Committee of the Chinese Center for Disease Control (CDC) and Prevention (approval number: 201519-A). All participants signed the informed consent document.

### 2.2. Data Collection and Variable Classifications

The medical examination was conducted by trained medical personnel who collect basic information (including age, nationality, education level, urban and rural household registration, income, alcohol consumption and smoking status) face-to-face according to standardized procedures. Weight and height were measured to the nearest 0.1 kg and 0.1 cm, respectively, without shoes and heavy clothing. Body mass index (BMI) was calculated as weight (kg) divided by height in square meters (m^2^). The waist circumference was measured with a tape measure with an accuracy of 0.1 cm. Systolic blood pressure (SBP)(mmHg) and diastolic blood pressure (DBP)(mmHg) were measured by Omron HBP-1300. The variety of variables was defined as follows. (1) Smoking status was categorized as yes or no, regardless of how often they smoked. If someone has quit smoking, they were considered a non-smoker. (2) Drinking status was categorized as yes or no in the last year. (3) The education level was divided into primary (primary school and below), medium (junior high school/senior high school), and advanced (junior college and above).

### 2.3. Laboratory Measurements

Overnight fasting blood samples were collected and placed in a vacuum blood collection vessel. They were then separated by centrifugation and stored in a freezer at −80 °C until analysis. This study used an enzymatic method in a biochemical analyzer (Hitachi 7600, Tokyo, Japan) to measure fasting plasma glucose (FPG), low-density lipoprotein cholesterol (LDL-C), total cholesterol (TC), triglyceride (TG), uric acid (UA), and high-density lipoprotein cholesterol (HDL-C). High-performance lipid chromatography (HPLC, Waters e2695, Milford, MA, USA) was used to measure the glycosylated hemoglobin (HbA1c). Inductively coupled plasma mass spectrometry (ICP-MS, PerkinElmer, NexION 350, Waltham, MA, USA) was used to measure the plasma Mg concentrations.

### 2.4. Definition of Prediabetes

According to the ADA in 2022 [16], prediabetes is a fasting plasma glucose (FPG) concentration of 5.6–6.9 mmol/L. Other than glucose, if the HbA1c level is 5.7–<6.5%, this should also be diagnosed as prediabetes.

### 2.5. Dietary and Nutrients Intake Assessment

The study used a validated food frequency questionnaire (FFQ) to assess dietary nutrient intake and dietary habits during the past 12 months [17]. There were 64 items on the questionnaire. It was mainly divided into several major categories such as staple food/beans/vegetables/mushrooms and algae/fruits/dairy/meat/seafood/eggs/other snacks/drinks/wine. The food eating frequency (daily/weekly/monthly/yearly) was used to calculate the daily eating weight, and then the data were defined as weight consumed in a single day. The individual daily intake of dietary energy(kcal), protein(g), fat(g), carbohydrate(g), calcium (Ca) (mg), Mg (mg), and food edible weight was calculated according to daily food and condiment consumption using the China Food Composition Table (2018) [18]. This study did not consider the use of Mg supplements because Mg supplementation is not popular in China. In addition, considering that dietary Mg intake may increase with energy, the total energy intake was adjusted by the residuals model [19].

### 2.6. Statistical Analysis

This study used SAS version 9.4 software (SAS Institute Inc., Cary, NC, USA) for all statistical data cleaning and analysis. The continuous variables were presented by the mean and standard deviation (SD), counts, and percentages for categorical variables. Their comparison used a *t*-test or chi-square test, respectively. We used multiple logistic regression to examine the relationship between Mg and prediabetes. The odds ratio (OR) values and 95% confidence intervals (95% CI) are compared with the lowest percentile in different models. We also used restricted cubic splines (RCS) to test for linearity and explore the shape of the dose-response effect between dietary Mg intake and prediabetes [20]. All *p*-values were two-sided, and when the *p*-value was lower than 0.05, it was considered statistically significant.

## 3. Results

### 3.1. Characteristics of Participants

The characteristics of 1981 childbearing Chinese women are displayed in Table 1. There were 1420 normal fasting glucose (NFG) people and 561 prediabetes cases. There was no significant difference between the two groups in the percentage of smoking status, drinking status, and nationality. Compared with the NFG group, the prediabetes group had a higher age (36.4 vs. 30.9), BMI (25.38 vs. 22.88), and a lower education level (23.6% vs. 40.5%) (*p* < 0.05). The participants in the prediabetes group had more western (36.4% vs. 32.3%) and city (53.7% vs. 42.5%) residences than the NFG group.

### 3.2. Clinical Characteristics of NFG and Prediabetes Group Participants

Clinical characteristics of 1981 childbearing Chinese women are displayed in Table 2. Compared with the NFG group participants, the prediabetes group participants had higher SBP, DBP, waist, weight, TG, TC, LDL-C, UA, FPG, and HbA1c levels. The prediabetes group had lower HDL-C and plasma Mg than the NFG group participants. All clinical characteristics showed significant differences (*p* < 0.05).

### 3.3. Mean Daily Nutrients Intake (Energy-Adjusted)

The mean daily nutrient intake is displayed in Table 3. There were no significant differences between the two groups in energy, protein, fat, and carbohydrate intake, and the ratio of calcium to magnesium (Ca/Mg). The prediabetes group participants had lower Ca and Mg intake. The difference was statistically significant (*p* < 0.05).

### 3.4. The Association between Dietary Magnesium and Prediabetes

This study’s generalized linear model showed the correlation factor between FPG and dietary Mg intake (Table 4). Overall, both before and after adjustments, dietary Mg was negatively associated with FPG. For every 100 mg of Mg intake, there was a 0.052 mmol/L (95% CI 0.019, 0.085) reduction in FPG (*p* = 0.002). The results of the multivariate logistic analysis of the associations between dietary Mg intake and prediabetes are summarized in Table 5. Lower ORs for prediabetes were associated with higher dietary Mg intake. Whether or not adjusted for age, education, district, residences, SBP, DBP, BMI, waist, TG, TC, HDL-C, LDL-C, UA, dietary calcium intake, the ORs (95% CI) for prediabetes in the Q3 and Q4 compared with the Q1 of dietary Mg were protective (0.63, 0.64), and the *p*-value was 0.020.

### 3.5. The Dose-Response Relationship between Dietary Mg Intake and Prediabetes

The spline regression model showed that the odds ratio of prediabetes decreased significantly with increasing dietary Mg at less than 410 mg/day and showed no significant decline thereafter (Figure 2).

## 4. Discussion

In the present study, the odds ratio of prediabetes decreased significantly with increasing dietary Mg intake (below 410 mg/day), and subsequently, no significant decrease was observed. To the best of our knowledge, this is the first such study undertaken among Chinese childbearing women.

First, this study observed the general negative association between dietary Mg and fasting blood glucose in Chinese childbearing women (β = −0.052, *p* = 0.002), and the prediabetes group had lower plasma Mg than the NFG group (*p* < 0.001). ARVIN-MIRMIRAN’s study [21] also found dietary Mg intake has an inverse association with the FPG (β = −0.08, *p* = 0.006). Besides this negative association, our study also certified the reduced odds ratio of higher Mg intake. Compared with the lowest intake quartile (<350.10 mg/day), the Q3 and Q4 (≥405.4 mg/day) quartiles reduced the odds of prediabetes by approximately 37%. WENSHUAI-LI’s study [22] used the data of The NHANES III follow-up US adult cohort, including 13489 participants aged 20 to 74 years old in the final analysis. They suggest that a higher intake of Mg may be associated with a lower risk of developing prediabetes. Their results showed that compared with the lowest intake quartile (≤192 mg/day), the highest quartile (>383 mg/day) reduced the odds of prediabetes by approximately 30% (OR = 0.72, 95% CI 0.53, 0.97, *p* = 0.02). ADELA-HRUBY [8] included 2582 community-dwelling participants aged 26–81 years old from the Framingham Heart Study (FHS) Offspring cohort. This study showed that higher Mg intake tended to be associated with lower FPG and insulin resistance. Over an average 6.9-year follow-up, they found that compared with the lowest Mg intake (236 mg/day), those with the highest intake (395 mg/day) had a 37% reduced odds ratio (OR = 0.63, 95% CI 0.45, 0.87) of prediabetes. GUERRERO-ROMERO conducted a cross-sectional study [23], which included 681 individuals who were 30 to 65 years old. Their dietary Mg intake was estimated according to a 24 h recall questionnaire. Their results also showed that hypomagnesemia is strongly associated with prediabetes (OR 2.19, 95% CI 1.1–7.0). This negative correlation between Mg intake and prediabetes is biologically viable and can be explained in part by the fact that Mg plays an important role in glucose and insulin metabolism, mainly through the effect of tyrosine kinase activity, which transfers phosphate from ATP to protein [24,25].

Second, this study explored the dose-response relationship between dietary Mg and blood glucose. Existing dose-response studies are almost based on people with T2DM, and there are few studies on prediabetes. This study shows that for every 100 mg of Mg intake, there was a 0.052 mmol/L reduction in FPG (*p* =0.002) (Table 5). CASTELLANOS-GUTIERREZ [26] included 1573 subjects from the 2012 Mexican Nutrition Survey. Data were locally representative. In this study, the relationship between Mg intake and blood glucose was only observed in women. They found that in women with normal glucose concentrations, an increase of 10 mg of magnesium per 1000 kcal/day resulted in an average decrease in serum glucose of 0.59 mg/dL (95% CI: −1.08, −0.09) (equal to 0.033 mmol/L). PAULA-NASCIMENTO’s research [27] used hierarchical grouping analysis to assess the association between the dietary intake of serum potassium, zinc, Mg, calcium, and glycemic indexes in 95 T2DM individuals. Their results showed a 0.7% reduction in %HbA1c per 100 mg of magnesium intake. XIN FANG [28] conducted a meta-analysis of prospective cohort studies of dietary Mg intake and the risk of developing T2DM. Their findings also showed an 8–13% reduction in T2DM incidence for each 100 mg/day increase in dietary Mg intake, after adjusting for age and BMI.

Third, in addition to the dose-response association between dietary Mg intake and blood glucose, this study also investigated the threshold effect relationship between them using the restricted cubic spline (RCS) method. The research on the threshold effect is not sufficient, especially in prediabetes. Almost all studies about the threshold research of dietary Mg intake, and T2DM or metabolic syndrome (MetS). One more thing I have to mention is that they used a different method. Like us, JIAO-YING et al. [11] used the data from China Health and Nutrition Survey (CHNS), proposing that when Mg intake is below 280 mg/d, the risk of metabolic syndrome (MetS) and its components are significantly reduced with increasing Mg intake. Their results were lower than ours. XU-TIAN [29] also used the RCS method to assess the dose-response association between dietary Mg intake and T2DM. Their fitting curve showed that 300 mg/day of Mg intake is a basic level for its effect against T2DM. However, their research was a meta-analysis of 15 prospective cohort studies, which is comprehensive but lacks specificity. BO-MA’s study [30] used the Bayesian method to evaluate the Mg threshold. The dietary intake of 1080 participants was assessed by FFQ. The threshold model found a statistically significant association with insulin sensitivity when Mg intake was less than 325 mg/day. That is lower than 410 mg/day, and this may be due to differences in the population, the end point of research, or statistical methods used. Except for the research on the threshold of dietary Mg intake, the GUERRERO-ROMERO 10-year follow-up study [31] showed an increased risk of IFG in patients with blood Mg ≤ 0.5 mmol/L and a decreased risk of IGT, IFG+IGT, and T2DM in patients with serum Mg concentrations ≥1.05 mmol/L. In terms of methodology, we still need more studies to discuss together. In this work, the prediabetes group had higher BMI and were older, which is consistent with CHEN and FANG’s research [6,7]; this may be because obesity and older age are risk factors for prediabetes. Therefore, our dose-response model and threshold effect model were corrected for BMI and age.

Regarding the investigated method of dietary Mg intake in the current study, we chose the validated FFQ, which was widely used in other studies [8,21,32,33]. XIN FANG [28]’s meta-analysis of prospective cohort studies included 25 cohorts of 637922 individuals and 26828 T2DM cases. All these cohort studies used FFQs to evaluate dietary Mg intake. The median Mg intake for the different dose groups ranged from 115 to 478 mg/day. Mg is widely found in whole grains and other unprocessed foods, such as nuts and green leafy vegetables [34]. This study’s FFQ involved the vast majority of food types and sources and used the China food composition table to calculate dietary Mg intake. Another dietary survey method is a three-day 24-h record/recall [35]. WENSHUAI-LI’s study [22] used NHANES’ single 24-h dietary recall method to derive the mean intake of 5815 prediabetes cases (51.3 years old, 44.6% female). These studies yielded different conclusions due to differences in overall design and methodology. To control for confounding factors, we used the residual method to analyze Mg intakes to eliminate the effect of energy.

This study has several strengths. First, the samples were a random sample selection from CNHS 2015, which had a complete quality control system and was nationally representative. Second, our focus on preventable prediabetes has greater public health implications. Third, we used both criteria (FPG and HbA1c) to make the included prediabetic population more representative. FPG represents blood glucose levels at a single point, and HbA1c represents the previous 3 months’ glucose levels. We comprehensively analyzed the relationship between dietary Mg intake and prediabetes, dose-effect, and threshold-effect relationship. Lastly, In the logistic regression and RCS model, we considered several covariates to obtain results that were as close to reality as possible.

Nevertheless, our study also has several limitations which need to be acknowledged. We used a validated questionnaire to assess the dietary intake of Mg during the last year, which may also cause recall bias. We only measured FPG and HbA1c, not OGTT and insulin, so we could not determine whether participants’ glucose tolerance was impaired or in an insulin resistance state. Lastly, our findings cannot establish a causal relationship between dietary Mg intake and the risk of prediabetes among Chinese childbearing women. Further work is required to resolve the extent and strength of the relationship between dietary Mg and prediabetes among Chinese childbearing women. Thus, a more randomized controlled trial about dietary Mg intake and prediabetes would be desirable to establish a causal relationship.

## 5. Conclusions

In this study, we found a negative association between dietary magnesium intake and prediabetes in childbearing Chinese women. The risk of prediabetes was reduced with increasing dietary magnesium intake, and the threshold value was 410 mg/day. This suggests that childbearing women also need to pay attention to their magnesium status.

## Figures and Tables

**Figure 1 nutrients-14-04580-f001:**
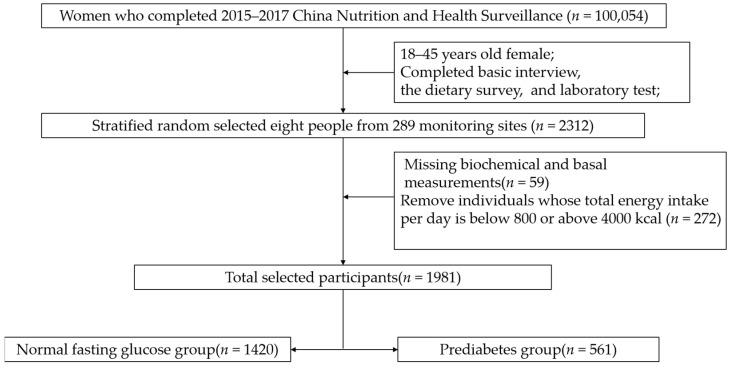
The flowchart of the selection of cases and controls.

**Figure 2 nutrients-14-04580-f002:**
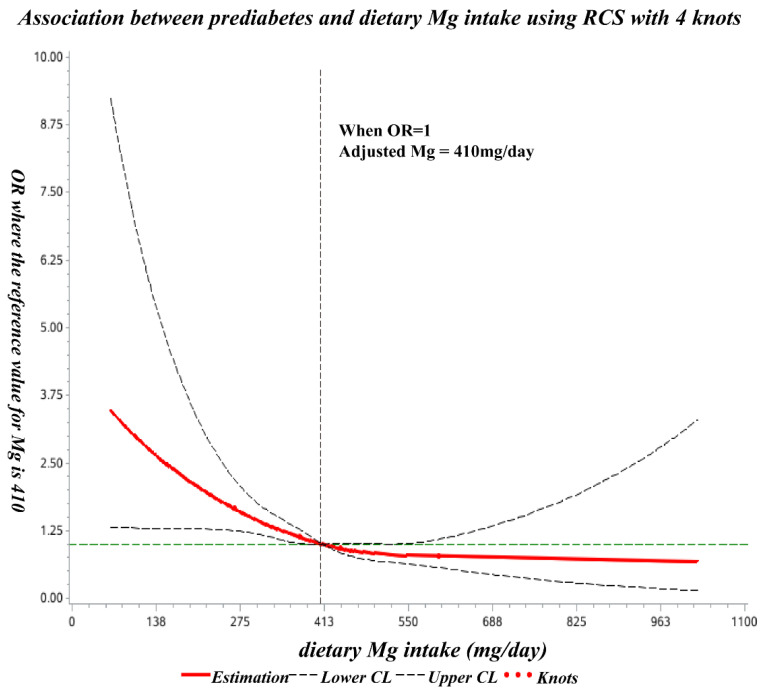
Dose-response relationship of dietary Mg intake with prediabetes OR. The ORs (solid lines) and 95% CIs (dashed lines) of dietary magnesium intake are indicated by straight lines based on restrictive cubic splines (RCS). Age, education, region, residence, SBP, DBP, BMI, waist circumference, TG, TC, HDL-C, LDL-C, UA, and dietary calcium intake were adjusted.

**Table 1 nutrients-14-04580-t001:** Basic Characteristics of NFG and Prediabetes Participants.

Characteristics	Total (*n* = 1981)	NFG (*n* = 1420)	Prediabetes (*n* = 561)	*p*-Value
	Mean ± SD/*n*	Mean ± SD/*n* (%)	Mean ± SD/*n* (%)	
Age(years)	32.4 ± 7.8	30.9 ± 7.5	36.4 ± 6.9	<0.001
Age group				<0.001
18–<26	551	494 (34.80%)	57 (10.20%)	
26–<36	671	498 (35.10%)	173 (30.80%)	
36–<45	759	428 (30.10%)	331 (59.00%)	
BMI (kg/m^2^)	23.59 ± 3.79	22.88 ± 3.53	25.38 ± 3.83	<0.001
BMI Group				<0.001
<18.5	134	125 (8.80%)	9 (1.60%)	
18.5–<24	1053	844 (59.40%)	209 (37.30%)	
24–<28	530	317 (22.30%)	213 (38.00%)	
28-	264	134 (9.40%)	130 (23.20%)	
Nationality				0.626
Han	1745	1254 (88.30%)	491 (87.50%)	
Ethnic minorities	236	166 (11.70%)	70 (12.50%)	
Education				<0.001
Primary	562	335 (23.60%)	227 (40.50%)	
Medium	1120	834 (58.70%)	286 (51.00%)	
Advanced	299	251 (17.70%)	48 (8.60%)	
District				0.034
Eastern	651	459 (32.30%)	192 (34.20%)	
Central	668	503 (35.40%)	165 (29.40%)	
Western	662	458 (32.30%)	204 (36.40%)	
Residences				<0.001
City	905	604 (42.50%)	301 (53.70%)	
Rural area	1076	816 (57.50%)	260 (46.30%)	
Smoke				0.202
yes	24	20 (1.40%)	4 (0.70%)	
no	1957	1400 (98.60%)	557 (99.30%)	
Drink				0.374
yes	414	304 (21.40%)	110 (19.60%)	
no	1567	1116 (78.60%)	451 (80.40%)	

Continuous variables are shown as Mean ± SD, and categorical variables are shown as percentages.

**Table 2 nutrients-14-04580-t002:** Clinical characteristics of NFG and Prediabetes Participants (Mean ± SD).

Index	Total (*n* = 1981)	NFG (*n* = 1420)	Prediabetes (*n* = 561)	*p*-Value
SBP (mmHg)	118.89 ± 15.03	116.17 ± 13.45	125.76 ± 16.58	<0.001
DBP (mmHg)	72.85 ± 9.92	71.29 ± 9.13	76.79 ± 10.71	<0.001
Waist (cm)	77.84 ± 9.82	76.14 ± 9.34	82.14 ± 9.70	<0.001
Height (cm)	156.98 ± 6.16	157.27 ± 6.15	156.27 ± 6.16	<0.001
Weight (kg)	58.19 ± 10.3	56.67 ± 9.81	62.04 ± 10.51	<0.001
TG (mmol/L)	1.20 ± 0.90	1.07 ± 0.74	1.54 ± 1.16	<0.001
TC (mmol/L)	4.38 ± 0.88	4.26 ± 0.83	4.68 ± 0.93	<0.001
LDL-C (mmol/L)	2.62 ± 0.77	2.51 ± 0.73	2.92 ± 0.81	<0.001
HDL-C (mmol/L)	1.30 ± 0.30	1.32 ± 0.29	1.25 ± 0.31	<0.001
UA (µmol/L)	265.16 ± 68.74	260.45 ± 66.16	277.08 ± 73.59	<0.001
FPG (mmol/L)	5.21 ± 0.80	4.79 ± 0.45	6.28 ± 0.39	<0.001
HbA1c (%)	4.79 ± 0.52	4.69 ± 0.46	5.03 ± 0.60	<0.001
Mg (mmol/L)	0.87 ± 0.08	0.88 ± 0.08	0.86 ± 0.09	<0.001

Abbreviation: SBP, systolic blood pressure; DBP, diastolic blood pressure; Waist, waist circumference; TG, triglycerides; TC, total cholesterol; LDL-C, low-density lipoprotein cholesterol; HDL-C, high-density lipoprotein cholesterol; UA, uric acid; FPG, fasting plasma glucose; HbA1c, Glycosylated hemoglobin; Mg, magnesium.

**Table 3 nutrients-14-04580-t003:** Mean daily nutrient intake (energy-adjusted) (Mean ± SD).

Nutrients	Total (*n* = 1981)	NFG (*n* = 1420)	Prediabetes (*n* = 561)	*p*-Value
Energy(kcal)	1955.85 ± 749.68	1939.24 ± 752.84	1997.89 ± 740.62	0.117
Protein(g)	60.03 ± 15.98	60.36 ± 16.37	59.18 ± 14.92	0.138
Fat(g)	25.78 ± 20.32	26.11 ± 20.50	24.95 ± 19.85	0.256
Carbohydrate(g)	413.19 ± 66.84	414.39 ± 68.34	410.16 ± 62.85	0.205
Calcium(mg)	372.01 ± 285.81	381.17 ± 292.46	348.84 ± 267.10	0.023
Magnesium(mg)	413.66 ± 102.12	419.21 ± 102.01	399.61 ± 101.15	<0.001
Ca/Mg ratio	0.91 ± 0.68	0.91 ± 0.67	0.89 ± 0.73	0.552

**Table 4 nutrients-14-04580-t004:** A generalized linear model of the associations between dietary magnesium intake and fasting plasma glucose mean daily nutrient intake (energy-adjusted) (Mean ± SD).

Model	β (Per 100 mg of Mg Intake)	95%CI		*p*-Value
		Lower	Upper	
Model 1	−0.056	−0.090	−0.021	0.001
Model 2	−0.049	−0.082	−0.016	0.004
Model 3	−0.052	−0.085	−0.019	0.002

Model 1 unadjusted; Model 2 added Age, Education, District, and Residences; Model 3 further adjusted SBP, DBP, Waist, BMI, TG, TC, HDL-C, LDL-C, UA, and Dietary calcium intake.

**Table 5 nutrients-14-04580-t005:** Multivariable-adjusted relationship between magnesium intake (mg/day) and prediabetes risk.

Model	Q1	Q2	Q3	Q4	*p*-Value
	<350.10	350.10–<405.41	405.41–<466.80	>466.89	
(NFG/Prediabetes)	(332/163)	(343/153)	(378/117)	(367/128)	-
Model 1	1 (ref)	0.91 (0.70,1.19)	0.63 (0.48,0.83)	0.71 (0.54,0.94)	0.016
Model 2	1 (ref)	0.85 (0.63,1.13)	0.63 (0.46,0.85)	0.71 (0.52,0.96)	0.004
Model 3	1 (ref)	0.81 (0.60,1.11)	0.63 (0.46,0.88)	0.64 (0.46,0.90)	0.020

Model 1 unadjusted; Model 2 added Age, Education, District, and Residences; Model 3 further adjusted SBP, DBP, Waist, BMI, TG, TC, HDL-C, LDL-C, UA, and Dietary calcium intake.

## Data Availability

Not applicable.

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
