# Peer review of "The Association of Dietary Magnesium and Prediabetes in Childbearing Chinese Women: Results from China Nutrition and Health Surveillance (2015–2017)"

_nutrients, 2022, doi:10.3390/nu14214580_

Round 1

Reviewer 1 Report

The title reports the association between magnesium levels and prediabetes in pregnant women.

You report reduced dietary magnesium intake and lower blood magnesium levels in women with prediabetes than in those with normal blood sugar.

However, it is curious how women with prediabetes are also those with higher BMI and older age.

There are no comments on weight and age in the discussion, the association with prediabetes is made only with the intake or magnesium levels.

There is probably no effect, however, in the patient table, age and body weight (variables that may be related to prediabetes) are highlighted as significant between the 2 groups.

I think that bringing this data into the discussion can be useful and underline the importance of magnesium

-----------------------------

If possible, it would be preferable to summarize the patients and methods in a flow-chart, to make the work carried out clearer to the reader.

Reviewer 2 Report

Thank you for the opportunity to review the manuscript, “The Association of Dietary Magnesium and Prediabetes in 2

Childbearing Chinese Women: Results from China Nutrition 3 and Health Surveillance (2015-2017),” submitted to Nutrients. 

The authors have conducted a nice case-control exploration of the association between dietary magnesium intake and pre-diabetes in Chinese childbearing women. In general the statistical models appear to be accurate and the results justified by the data and their methods period their results appear to agree with other investigators as well which indicate a small but significant decrease in pre-diabetes with increasing magnesium intake particularly towards the lower half of magnesium intake distribution.  My only concern with the present manuscript is that in table 2 some of the P values seem suspicious for typographical or mathematical error, specifically height and chemical measures seem to have overlapping standard deviations, yet are accorded p<0.001. 

Reviewer 3 Report

Dear authors, 

I read you mnauscript competeely. I dont have much to comments since this manuscript is written in a very good form. However, there are minor points I would like authors  to work on

- the ethics section is completely missing from material and methods section. I strongly recommend authors to include this section with approval numbers and NTC numbers (if any)

-Dony use the word "I" or "us" in the text. nstead use "in this study, or in this work". 

-Improve yout conclusion with more findings in this work. 

-add comments about the inclusion of MG in diets in different food cultures and societies. 

Thanks.  
